# Leaf Gas Exchange of Tomato Depends on Abscisic Acid and Jasmonic Acid in Response to Neighboring Plants under Different Soil Nitrogen Regimes

**DOI:** 10.3390/plants9121674

**Published:** 2020-11-29

**Authors:** Shuang Li, Abdoul Kader Mounkaila Hamani, Zhuanyun Si, Yueping Liang, Yang Gao, Aiwang Duan

**Affiliations:** 1Key Laboratory of Crop Water Use and Regulation, Ministry of Agriculture and Rural Affairs, Farmland Irrigation Research Institute, Chinese Academy of Agricultural Sciences, Xinxiang 453002, China; li_shuang0712@163.com (S.L.); m_abdoulkader@yahoo.com (A.K.M.H.); sunbirdszy@163.com (Z.S.); yueping0520@163.com (Y.L.); 2Graduate School of Chinese Academy of Agricultural Sciences (GSCAAS), Beijing 100081, China

**Keywords:** neighboring competition, limited nitrogen application, stomatal conductance, photosynthesis, abscisic acid, jasmonic acid

## Abstract

High planting density and nitrogen shortage are two important limiting factors for crop yield. Phytohormones, abscisic acid (ABA), and jasmonic acid (JA), play important roles in plant growth. A pot experiment was conducted to reveal the role of ABA and JA in regulating leaf gas exchange and growth in response to the neighborhood of plants under different nitrogen regimes. The experiment included two factors: two planting densities per pot (a single plant or four competing plants) and two N application levels per pot (1 and 15 mmol·L^−1^). Compared to when a single plant was grown per pot, neighboring competition decreased stomatal conductance (g_s_), transpiration (T_r_) and net photosynthesis (P_n_). Shoot ABA and JA and the shoot-to-root ratio increased in response to neighbors. Both g_s_ and P_n_ were negatively related to shoot ABA and JA. In addition, N shortage stimulated the accumulation of ABA in roots, especially for competing plants, whereas root JA in competing plants did not increase in N15. Pearson’s correlation coefficient (*R^2^*) of g_s_ to ABA and g_s_ to JA was higher in N1 than in N15. As compared to the absolute value of slope of g_s_ to shoot ABA in N15, it increased in N1. Furthermore, the stomatal limitation and non-stomatal limitation of competing plants in N1 were much higher than in other treatments. It was concluded that the accumulations of ABA and JA in shoots play a coordinating role in regulating g_s_ and P_n_ in response to neighbors; N shortage could intensify the impact of competition on limiting carbon fixation and plant growth directly.

## 1. Introduction

Neighboring plant competition and soil mineral shortages resulting from an increasing planting density are common biotic and abiotic stresses, limiting crop growth and production [1,2]. Nitrogen (N), as one of the most essential macronutrients for plant growth and development, plays a significant role in carbon metabolism, protein synthesis, and plant hormone synthesis [3]. Plants may detect and interact with neighbors in different manners [4]. Recent data argue that phytohormones, such as ethylene and abscisic acid (ABA), play a crucial role in the modulation of many processes of plant development, including response to neighbors [5,6]. Transcriptomic analysis of the whole-genome expression showed that nutrient deficiency (mainly N) upregulated genes of ABA biosynthesis in competing *Arabidopsis* plants [7]. However, the combined effects of the two factors on endogenous hormones remain unclear, particularly on ABA. Plants growing in high planting canopies usually receive low-quality light (measured as low red to far-red R/FR ratio), an early warning sign of crowding [8]. At the morphological level, past studies have shown that plants’ response to a low R/FR ratio commonly exhibited an increase in stem and petiole elongation, more vertical leaf orientation, and apical dominance [5,9,10]. There is convincing evidence of the participation of hormones (ethylene, auxin and cytokinin) in the regulation of shoot and root growth [8,11,12]. Stomata, controlling the balance between the water loss by transpiration and CO_2_ assimilation, could rapidly respond to environment stimuli [13]. At the physiological level, neighboring competition resulted in a reduction of stomatal conductance (g_s_) and leaf transpiration rate (T_r_) in wild-type (*WT*) tomatoes with no decrease in leaf water status, whereas g_s_ and T_r_ did not decrease in an ABA-deficient tomato mutant [6]. Moreover, Arkhipova et al. [14] found that lettuce plants inoculated with *Bacillus subtilis*, strain IB-22, capable of producing cytokinin, did not reduce T_r_ in response to neighbors. These findings confirmed the involvement of plant hormones (ethylene, ABA, auxin and cytokinin) in regulating plant growth and stomatal movement in response to high planting densities. Jasmonic acid (JA) and its derivates, such as methyl jasmonate (MeJA), a natural endogenous growth regulator, are also known to modulate plant morphological, physiological, and biochemical processes in response to drought [15,16,17], cold [18] and salt stress [19,20]. Experiments with JA have been shown to decrease g_s_ and T_r_ in wild-type tomato and ABA-deficient mutants [21]. More recently, de Ollas et al. [22] also reported that stomatal sensitivity to drying was lower in an *def-1* self-grafted mutant (which fails to accumulate JA in shoots) plants as compared to the def-1 scions grafting with WT rootstocks. These results confirmed that a high production of JA was required to regulate g_s_ in response to stresses. However, the shade-inducing lowering of R/FR ratio has been shown to reduce the accumulation of JA [9,23]. Taken together, the closure of stomata relies on JA accumulation, while the dense canopy might reduce JA accumulation. It is thus still far from clear how the concentration of JA in shoots and roots changes in response to high planting densities. In addition, few studies have demonstrated that the role of JA inhibiting stomatal opening requires ABA, as JA interacts with ABA biosynthesis [24,25]. However, the interactions between JA and ABA in regulating g_s_ at an increased planting density are still unknown.

Accumulated evidence showed that N deficiency reduced carbon assimilation rate and plant growth [26,27,28]. Some studies have shown that N deficiency-induced low net photosynthesis rate (P_n_) was due to the reduced stomatal conductance and the photosynthetic capacity, such as chlorophyll index, nitrate reductase activity, and rubisco content [26,27]. The relationship between plant competitors might become more complicated under infertile soil conditions. However, there is little information about N deficiency’s effect on the relationship between hormones and leaf gas exchange under an increased planting density. 

The tomato (*Solanum lycopersicum* L.), one of the most important vegetable crops, is susceptible to neighboring competition in a soil environment with insufficient nitrogen. Therefore, the objectives of this research are (i) to analyze the effect of neighboring competition on JA content in roots and shoots under different soil nitrogen regimes, (ii) to assess the involvement of ABA and JA on leaf gas exchange in the responses of neighbors under different soil nitrogen regimes; (iii) and to clarify whether soil N deficiency increases the effect of neighboring competition on hormones, gas exchange, and plant growth. This work could provide new knowledge on the roles of ABA and JA in modifying tomato leaf gas exchange in response to neighbors under limited soil N conditions. 

## 2. Results

### 2.1. Leaf Water Status

As shown in Figure 1, no significant difference in shoot water potential (ψ_shoot_) between a group of four plants and singly grown plants was found in N15-and N1-treated plants. Two-way analysis of variance (ANOVA) results showed that competition, nitrogen, and their interaction did not affect ψ_shoot_, suggesting that the plant water status was unaffected by these two factors in this study. 

### 2.2. Plant Growth and Leaf Gas Exchange

Two-way ANOVA showed that competition, nitrogen, and their interaction could significantly impact the growth indexes (Table 1). Compared with a single tomato plant being grown in one pot (P1), competition from neighbors (P4) did not affect shoot length under N15 at the end of our experiment (*p* = 0.193), but significantly decreased the shoot length under N1 (*p* = 0.004). However, P4 statistically reduced the stem diameter and leaf area of tomato plants under both N15 and N1 applications. Generally speaking, with the presence of neighbors, these indexes were much lower in the N1-treated plants than in N15-treated plants. Similarly, the dry biomass of competing plants was 38.5% and 25.3% lower than for plants grown singly under N15 and N1, respectively (Appendix A). 

The net photosynthesis rate (P_n_), stomatal conductance (g_s_) and transpiration rate (T_r_) were significantly affected by competition and nitrogen application. P_n_ was suppressed by neighbor competition as compared to the singly grown plants (*p* = 0.002), and low nitrogen aggravated the suppression of P_n_ in grouped plants (*p* = 0.000) (Figure 2A). The response of g_s_ and T_r_ to competitors was similar to P_n_, and much lower under low N application (Figure 2B,C). Intercellular CO_2_ concentration (C_i_) in competing plants significantly increased in comparison to single plants under N15 conditions (*p* = 0.022), but significantly decreased as compared to single plants under N1 treatment (*p* = 0.04) (Figure 2D). 

### 2.3. Hormonal Response

Competition and the interaction among two factors significantly affected the changes of ABA in shoots, roots and the ratio of shoot-to-root ABA ratio (Table 2). As shown in Figure 3A, the presence of neighbors significantly increased ABA in shoots, which was almost three times higher in N15 application (*p* = 0.000) and 1.4 times higher in N1 application (*p* = 0.006), respectively. The concentration of shoot ABA in N15P4-treated plants was higher than in N1P4-treated plants (*p* = 0.009). Interestingly, the concentration of root ABA in N1-treated plants was higher than in N15-treated plants. As a result, competition induced an increase in the shoot-to-root ABA ratio in N15-treated plants (*p* = 0.014). In contrast, N shortage did not affect the ratio between single and grouped plants due to the remarkable root ABA accumulation (Figure 3A). Changes of JA in shoots, roots, and the shoot-to-root ratio were significantly affected by the neighboring competition, nitrogen, and their interaction (Table 2). Competition significantly increased the concentration of JA in shoots regardless of N treatments (*p* = 0.019 in N15 and *p* = 0.000 in N1, respectively), and soil N shortage resulted in almost 2.3-fold higher shoot JA as compared to that of a single plant (Figure 3B). The concentrations of JA in the roots of grouped plants did not increase in N15-treated plants, but increased dramatically in N1-treated plants in response to neighbors (*p* = 0.001). Therefore, competition significantly increased the shoot-to-root JA ratio in N15 and N1 plants (*p* = 0.012 in N15 and *p* = 0.000 in N1, respectively), and the ratio of JA in N1 was lower than in N15-treated plants. 

### 2.4. Correlations of g_s_, P_n_, Hormones, and Leaf N Content

In N15-treated plants, g_s_ negatively responded to the ABA in shoots, roots, and the shoot-to-root ratio, and the Pearson’s correlation coefficient (*R^2^*) increased in N1-treated plants (Table 3). Although there was no significant relationship between g_s_ and root JA or shoot-to-root JA ratio in N15-treated plants, this relationship significantly increased in N1-treated plants; the *p* values were both equal to 0.000. The correlations of P_n_ with ABA and JA were significant in N15-treated plants, and *R^2^* was higher in N1 plants. g_s_ decreased linearly with increasing shoot ABA concentration in response to neighboring competition in both N15 and N1 (Figure 4A). The absolute value of slope of g_s_ to ABA in N1-treated plants was higher than in N15-treated plants. g_s_ also decreased with increasing shoot JA as compared to a single plant. However, the slope of g_s_ to JA did not increase in N1-treated plants (Figure 4B). g_s_ increased linearly with the increase in leaf N content, and the *R^2^* was very high and significant at the *p* < 0.01 level, being 0.92 (Figure 4C). 

### 2.5. Correlations of Chlorophyll Index and Leaf nitrogen Content

The effects of competition and nitrogen on chlorophyll index (as measured by soil–plant analysis development, SPAD), were significant at the *p* < 0.01 level, whereas the interaction among these two factors was insignificant (Table 4). The SPAD of grouped plants decreased in both the N15-and N1-treated plants (Figure 5A). SPAD in N1-treated plants was much lower than in N15. Competition did not significantly affect the contribution of stomatal limitation (L_s_). In contrast, nitrogen and the interaction between these two factors could substantially affect L_s_ and the limit of non-stomatal conductance (L_ns_) (Table 4). The L_s_ in grouped plants was lower than in single plants per pot under N15 conditions (*p* = 0.022), but L_s_ increased in N1P4 as compared with the N1P1 treatment (Figure 5B) (*p* = 0.040). The L_ns_ was much higher in N1P4 than in other treatments. 

SPAD decreased linearly with the decreasing leaf N concentration in response to neighbors in both N15 and N1, and the slope was higher in N1 than in N15 (Figure 6A). The relationship of P_n_ to SPAD was similar to that of SPAD to leaf N concentration, and the slope increased in N1 as compared to N15 (Figure 6B). 

## 3. Discussion

There is a common consensus that plants increase the length of the stem and petiole to gain more light intensity in response to the presence of neighbors [11]. Likewise, in this study, the shoot length of grouped plants did not decrease in sufficient N supply condition (Table 1). These results suggested that plants might suffer from neighboring competition under an elevated planting density. However, N deficiency seriously impeded plant growth, especially for grouped plants. 

### 3.1. Effects of Neighbor Competing and Nitrogen Availability on Leaf Gas Exchange and Abscisic Acid (ABA)

In competing plants g_s_ and T_r_ decreased, but the shoot water potential was unaffected as all plants were well-watered (almost 70–75% total sand moisture capacity, TMC) during the experiment, so the leaf hydraulic signals might not drive the closure of stomata. Nevertheless, there was a negative relationship between g_s_ and ABA (Table 3). Previous studies have reported that competition-induced stomatal closure was associated with the accumulation of ABA in shoots due to the increased transport of ABA from root to shoot under the increased planting density [6,29,30]. Our data also supported this view because more ABA accumulated in shoots than in roots in competing plants, resulting in a 1.7-fold increase in the shoot-to-root ABA ratio (Figure 3A). The importance of root-to-shoot ABA transport was supported by data showing that enhanced root-to-shoot ABA transport decreased g_s_ in ABA-deficient *flacca* mutant scions grafted with WT rootstock compared to *flacca* rootstocks [30]. Experiments with fluridone, an inhibitor of ABA transport, showed that the absence of ABA accumulation in shoots prevented growth inhibition in competing plants [31]. However, the shoot-to-root ABA ratio did not increase in response to neighbors under N deficiency conditions and this ratio was much lower in N1-treated plants than in N15-treated plants, implying that low N availability was able to decrease ABA transport from root to shoot (Figure 3A). One of the reasons might be the lower T_r_ in N1-treated plants (Figure 2C). Transpiration pull was considered to be the main power for plants to transport water, hormones and other nutrients from the roots [32]. N supply was expected to have a major effect on leaf gas exchange, particularly on g_s_ and T_r_ [33,34]. Another reason may be more ABA being produced in the roots under N deficiency conditions. The results of more ABA accumulating in N-deprived plants were in agreement with the data of Oka et al. [35], who showed that low nitrogen availability promoted ABA accumulation. N deficiency has been shown to upregulate zeaxanthin epoxidase (TaZEP), a key ABA synthesis-related enzyme and increase the endogenous ABA content [36]. Interestingly, with no increase in ABA export from roots, g_s_ in nitrogen-deficient plants showed a fast decrease in response to the accumulation of ABA in shoots of competing plants (Figure 4A). Moreover, the relationship between g_s_ and ABA in shoots and roots, and the shoot-to-root ratio of N1-treated plants, was higher than that of N15-treated plants (Table 3). Thus, it is plausible that different mechanisms might exist in the response of g_s_ to competitors under low nitrogen conditions. One of the reasons for the high sensitivity of g_s_ to ABA under N deficiency might be that N deprivation alkalized the xylem sap, a moderator of the ABA signaling process [29]. The increased planting density increased the pH of the xylem sap and soil solution as more nitrate was absorbed from the same volume of soil [6,37]. Furthermore, it has been shown that g_s_ decreased with decreasing leaf N content [26,38,39]. A positive relationship between g_s_ and leaf N content was also observed in this study (Figure 4C), which suggested that N might have a direct influence on the stomatal movement (Figure 7). Although the mechanisms of N in the regulation of g_s_ are not investigated in depth in this study, it is plausible that N deficiency sensitizes g_s_ to ABA in response to neighbor competition (Figure 7). 

### 3.2. Effects of Neighbor Competing and Nitrogen Availability on Leaf Gas Exchange and Jasmonic Acid (JA) 

To understand the role of JA in response to neighbor competition, changes of JA in shoots and roots were also investigated in our experiment. We found that the concentration of JA in shoots increased in competing plants (Figure 3B). The data presented here were in agreement with previous studies showing that the concentration of JA in shoots was able to increase in response to water stress [17,40]. Shoot-to-root JA ratio increased simultaneously in grouped plants, implying that neighbors’ presence enabled increased JA transport from root to shoot (Figure 7). The importance of root to shoot JA transport has been stressed by De Ollas et al. [19], who found that attenuated root to shoot JA transport improved the leaf water status under soil drying conditions. However, there is a knowledge gap about the mechanisms involved in the control of JA allocation under increasing density. Future research should focus on the effects of neighboring competition on the root to shoot JA transport. By contrast, some studies showed that low R/FR ratio light, an early warning sign of crowding, would inhibit the expression of JA-induced genes and promote disease symptoms in Arabidopsis [41] and tomatoes [42]. Although we did not measure the effect of JA on plant disease symptoms, different results might be attributable to different plant materials, cultivars and growth conditions. At the same time, the present work showed that the g_s_ of the grouped plants decreased with increasing shoot JA, indicating that g_s_ negatively responded to JA (Table 3, Figure 4B). Previous studies have shown that JA-induced stomatal closure was closely related to cytoplasmic alkalinization in guard cells, the production of reactive oxygen species (ROS) and NO, activation of K^+^-efflux, and slowing anion channels [43,44,45]. Therefore, the role of JA in inducing stomatal closure response to neighbors was similar to that of ABA, indicating that there might be a synergistic rather than an antagonistic interaction between ABA and JA in regulating stomatal movement in response to biotic stress (Figure 7). Suhita et al. [39] reported that the stomata of ABA-insensitive mutants were able to close in response to the exogenous application of JA. In addition, the sensitivity of g_s_ to drying decreased in *def-1* self-grafts (which fail to accumulate JA in shoots during drying) as compared to *def-1* scions grafted onto *WT* rootstocks [19]. Since the role of JA in regulating ABA biosynthesis has been addressed [21], the coordinated action of ABA and JA may be due to their converged signal transduction pathways at the level of intracellular Ca^2+^ modulation [45]. However, our data cannot fully explain this interaction. In order to reveal the interaction of ABA and JA, grafted mutant plants should be used in the future. Moreover, as shown in Table 3, tomato seedlings displayed a negative correlation between JA in shoots and P_n_, such as ‒0.791 and ‒0.977 in N15 and N1 conditions, respectively. The negative role of JA on photosynthesis might be due to the down-regulation of the rubisco enzyme by JA [46]. These results suggest the important role of JA accumulation in the modulation of leaf gas exchange to adapt to competitors. N deficiency decreased the concentration of JA in shoots and shoot-to-root ratio as compared to the N15 treatment (Figure 3B), in agreement with the findings of Jang et al. [47], who found that elevated N application increased JA concentrations in shoots of rice cultivars as compared with no N supply. However, unlike the g_s_ to shoot ABA, the slope of g_s_ to shoot JA of grouped plants under N deficiency did not change as compared to in N-sufficient conditions (Figure 4B), suggesting that N would not change the sensitivity of g_s_ to JA in response to neighbors. Although the insensitivity of g_s_ to JA in the face of N shortage was not investigated in depth in this study, grouped plants had the lowest g_s_ under N deficiency conditions. 

### 3.3. Effects of Neighbor Competing and Nitrogen Availability on P_n_


Under the N sufficiency condition, the limitation of g_s_ to photosynthesis of grouped plants was lower than of plants grown singly. However, grouped plants grown in N deficiency conditions showed higher L_s_ values. Thus, one reason for the decline in photosynthesis of grouped plants in N insufficiency might be the closure of stomata due to the low CO_2_ supply for carboxylation [48]. The N1-treated plants had a higher L_ns_ value, especially for grouped plants, indicating that photosynthesis was simultaneously limited by non-stomatal factors, such as the significant decrease of SPAD accompanied by a decline in leaf N content in this study (Figure 6A). Moreover, the slope of SPAD to P_n_ in N1 was higher than in N15-treated plants, suggesting that P_n_ in N-deficiency plants was more sensitive to SPAD (Figure 6B). Our findings were consistent with those of other studies [26,49], which found that the N supply had a positive effect on SPAD values, as N was a major source for chlorophyll synthesis [50]. Hence, it was not surprising that plant growth parameters such as the stem diameter and dry biomass of group plants were severely inhibited by low nitrogen availability (Table 1, Appendix A). 

## 4. Materials and Methods 

### 4.1. Plant Materials and Experimental Design

The indoor experiment was conducted in a controlled environment room at the experimental station of the Institute of Farmland Irrigation (35°18′ N, 113°54′ E; 81 m altitude). Tomato seeds (*Solanum lycopersicum* L., cv. Helan108) were germinated in a nursery seedling plate with substrate (sphagnum peat, Pindstrup Mosebrug A/S, Ryomgaard, Denmark). When tomato seedlings were 16 days old, those with two true leaves were transplanted into pots (height 15 cm, diameter 16 cm) and grown in a controlled environment room with a 14-h photoperiod (6:00 to 20:00), provided by helium and sodium lamps with 400 µmol·m^−2^·s^−1^ photosynthetic active radiation (PAR) at 25–30 °C with 45–55% relative air humidity. Each pot was filled with 3.5 kg sterile sand. Two experimental factors with two levels (i.e., planting density and nitrogen rate) were designed in this study. The planting density levels were maintained by transferring a single plant (P1) or four plants (P4) into one pot; the two levels of nitrogen rate were 1 mmol·L^−1^ (N1) and 15 mmol·L^−1^ (N15), respectively. Chloride was used as a counter-ion to regulate the K^+^ in the N1 solution. KOH solution was used to adjust the pH to 6.0 ± 0.2 of both in the N15 and N1 solutions. After transplanting, each pot was watered with 600 mL of 1/2 complete Hoagland solution (1 and 15 mmol·L^−1^, respectively). From 2 DAT (days after transplanting, DAT) up to harvest at 14 DAT, all pots were weighed in the morning and irrigated with 20 mL half-strength Hoagland solution depending on the nitrogen treatments, with additional distilled water to reach a level of 70–75% total sand moisture capacity (TMC). To prevent soil evaporation, pots were covered with aluminum foil. 

### 4.2. Measurement of Plant Development, SPAD and Leaf N Content

At 14 DAT, plant growth parameters like shoot length, stem diameter, biomass and leaf area were measured with six replications of each treatment. The shoot length of tomato seedlings was measured using a ruler. A vernier caliper was used to measure the stem diameter 1 cm from cotyledon. Leaf area was recorded by the LI-3000C Area Meter (LI-COR Inc., Lincoln, NE, USA). The middle part of the fully expanded leaf was chosen to measure the chlorophyll index (SPAD) using a Minolta SPAD 502 chlorophyll meter (Minolta Camera Co., Ltd., Osaka, Japan). The dry weight was obtained after oven drying at 75 °C for 24 h. Leaf N concentrations were determined with a continuous flow autoanalyzer (Bran + Luebbe, Hamburg, Germany). 

### 4.3. Determination of Leaf Gas Exchange Parameters

Leaf gas exchange parameters, including P_n_, g_s_, T_r_ and the intercellular CO_2_ concentration (C_i_), were measured, with six replicates, on the same leaves used for SPAD measurement, using a LI-6400XT portable photosynthesis system (LI-COR Inc. Lincoln, NE, USA) equipped with a red and blue light source chamber. Each measurement was taken between 9:00 and 13:00. On the leaf chamber, PAR was set to 400 μmol·m^−2^·s^−1^, which was near the environmental light intensity; the sample CO_2_ concentration was maintained at 400 μmol·mol^−1^ using a CO_2_ mixer. The block leaf temperature was 28 °C. 

The limitation of g_s_ to photosynthesis was calculated as follows, L_s_ = 1 − C_i_/C_a_ [51], and the limit of non-stomatal conductance was then calculated as L_ns_ = C_i_/g_s_, where C_a_ is the ambient CO_2_ concentration (C_a_ = 400 μmol·m^−2^·s^−1^).

### 4.4. Root and Shoot Sap Collection 

After the measurement of leaf gas exchange, tomato seedlings were cut off at 1–2 cm above the sand surface. Then, root sap was collected from the cut surface and placed into a centrifuge tube in liquid nitrogen. Plant shoots were placed into the pressure chamber Model 3115 Pressure Extractor (Soil Moisture Equipment Corp., Santa Barbara, CA, USA) with the stem stump outside the chamber. After cleaning the cut surface with pure water, the pressure was increased slowly, and we noted the pressure when the first drop of solution occurred. The negative pressure was regarded as the shoot water potential. To collect the sap solution, 0.3‒0.5 MPa pressure was continuously applied to the shoot. Then, 2-mL samples of sap collected from some tomatoes of each treatment were immediately placed into centrifuge tubes in liquid nitrogen. Sap solutions were stored at −80 °C for hormone analysis. 

### 4.5. ABA and JA Measurement 

ABA and JA concentrations were determined according to the method described by You et al. [52] with little modification. First, 2 mL of extracted sap sample were placed into a microcentrifuge tube containing 5 mL extraction buffer composed of isopropanol/hydrochloric acid and 8 μL 1 μg mL^−1^ deuterated internal standard solutions. Then, 10 mL dichloromethane were added and sap samples were shaken at 4 °C for 30 min and then centrifuged at 13,000× *g* for 5 min at 4 °C. The supernatant was carefully removed and the lower organic phase was dried by N_2_ in shade conditions and dissolved in 400 μL methanol (containing 0.1% methane acid), then filtered with a 0.22-mm filter membrane. The purified sample was then subjected to high-performance liquid chromatography‒tandem mass spectrometry (HPLC-MS/MS), fitted with a POROSHELL120 SB-C18 (Agilent Technologies Inc., Santa Clara, CA, USA) column (2.1 mm × 150 mm; 2.7 mm), at 30 °C. The solvent gradient used was 100% A (99.9% methanol: 0.1% CHOOH) to 100% B (99.9% H_2_O: 0.1% CHOOH) over 15 min. The injection volume was 2 μL. MS conditions were as follows: the spray voltage was 4500 V; the pressure of the air curtain, nebulizer, and aux gas were 15, 65, and 70 psi, respectively; and the atomizing temperature was 400 °C. The ABA and JA concentrations were calculated with reference to the peak area of the deuterated internal standard.

### 4.6. Statistical Analysis 

Statistical differences among different treatments were assessed by two–way analysis of variance using SPSS 16.0 (IBM SPSS Statistics, Chicago, IL, USA). Multiple comparisons between all treatments were analyzed using a Duncan’s test. All data are presented as the mean ± standard error (SE). Different letters were considered significant at the *p* < 0.05 and *p* < 0.01 level. Data fitting and graphical presentation were carried out in Origin-Pro 2017 (Origin Lab, Northampton, MA, USA). The general linear regression model was used to fit the relationships between parameters under different N application levels.

## 5. Conclusions

Endogenous JA in the shoots and shoot-to-root ratio are affected by the presence of neighbors, JA works synergistically with ABA in the regulation of stomatal movement and carbon fixation under elevated planting density. Negative correlations between JA or ABA and both g_s_ and P_n_ increase under N deficiency conditions. At the same time, a soil nutrient shortage somewhat increased the sensitivity of g_s_ to ABA in response to neighbors, whereas N shortage does not increase the g_s_ to JA in response to neighbors. Stomatal limitations and non-stomatal limitations contribute to the decrease of P_n_ in N shortage conditions. Future studies should focus on how neighbors would affect the root to shoot JA transport or the interaction between ABA and JA. However, our results shed new light on the important role of ABA, JA, and their orchestrated signaling in the regulation of plant responses to neighbor competition in a complex growth environment. 

## Figures and Tables

**Figure 1 plants-09-01674-f001:**
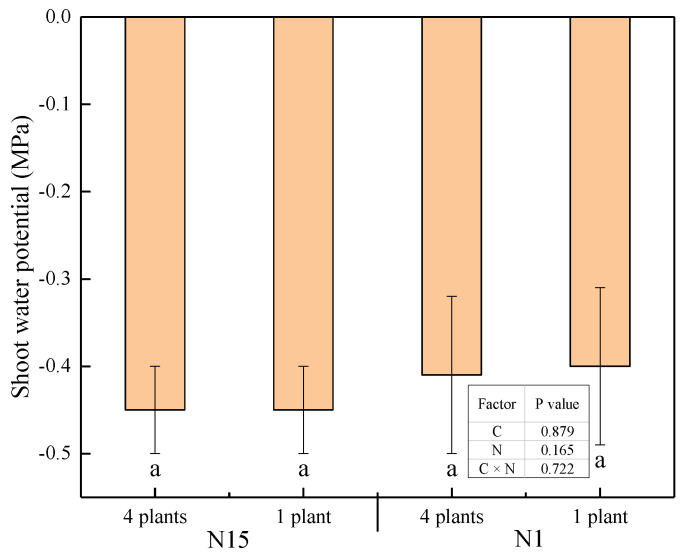
Effects of competition and nitrogen supply on shoot water potential (ψ_shoot_) 14 days after transplanting. Mean values and error bars of standard error (SE) are presented (*n* = 6). N15 and N1 mean 15 mmol·L^−1^ and 1 mmol·L^−1^ supply, respectively; 4 plants and 1 plant mean four grouped plants or single plants grown in one pot, respectively. The table summarizes the significance (*p*-value) of competition treatment (C), nitrogen treatment (N), and their interaction (C × N) by two-way analysis of variance (ANOVA), Duncan’s multiple-range test.

**Figure 2 plants-09-01674-f002:**
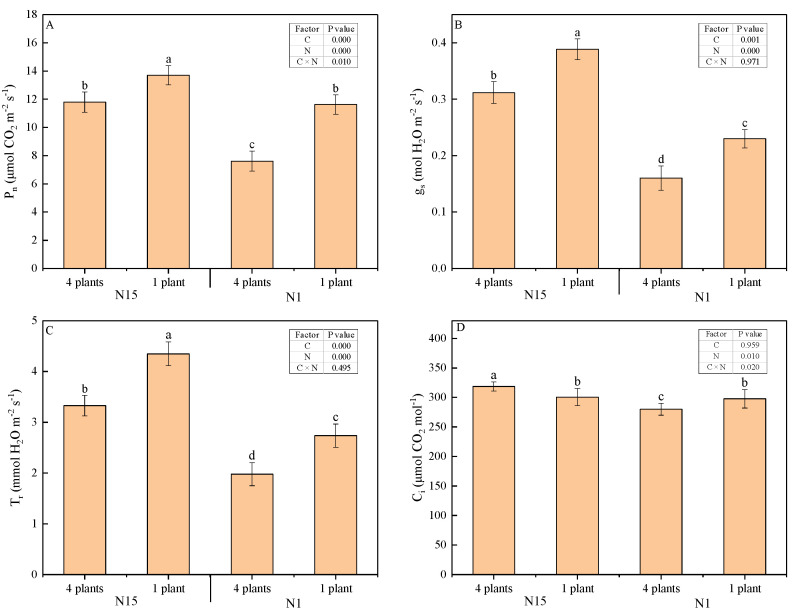
Net photosynthetic rate (**A**), stomatal conductance (**B**), transpiration rate (**C**), and intercellular CO_2_ concentration (**D**) in response to neighbors in two nitrogen applications 14 days after transplanting. Mean values and SE (*n* = 6) are presented. The table summarizes the significance (*p*-value) of competition treatment (C), nitrogen treatment (N), and their interaction (C × N) by two-way ANOVA, Duncan’s multiple-range test. Different letters indicate a significant difference at the *p* < 0.05 level by Duncan’s multiple-range test.

**Figure 3 plants-09-01674-f003:**
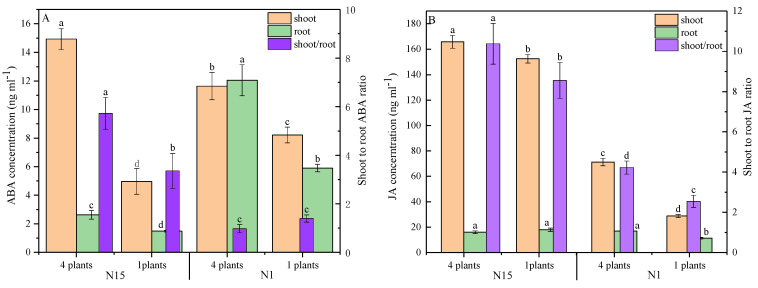
Changes of ABA (**A**), JA (**B**), and shoot-to-root ratio in response to neighbors in two nitrogen applications 14 days after transplanting. Mean values and SE (*n* = 3) are presented. Different letters indicate a significant difference at the *p* < 0.05 or *p* < 0.01 level by Duncan’s multiple-range test.

**Figure 4 plants-09-01674-f004:**
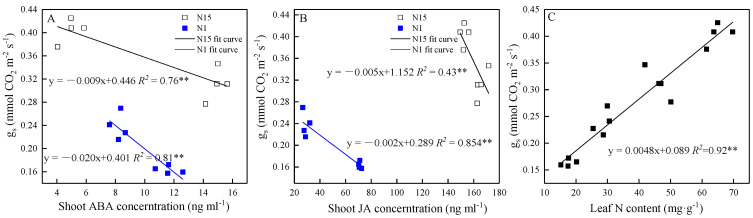
Relationships of stomatal conductance (g_s_) to shoot ABA concentration (**A**), g_s_ to shoot JA concentration (**B**), under N15 and N1 treatments, and the g_s_ to leaf N content under all treatments (**C**). In (**A**) and (**B**), the open squares indicate plants under N15 treatment and closed squares indicate plants under N1 treatment. A fit curve is presented with the fitted equation (*n* = 8 or 16); ** indicates that the regression lines are statistically significant at the *p* < 0.01 level.

**Figure 5 plants-09-01674-f005:**
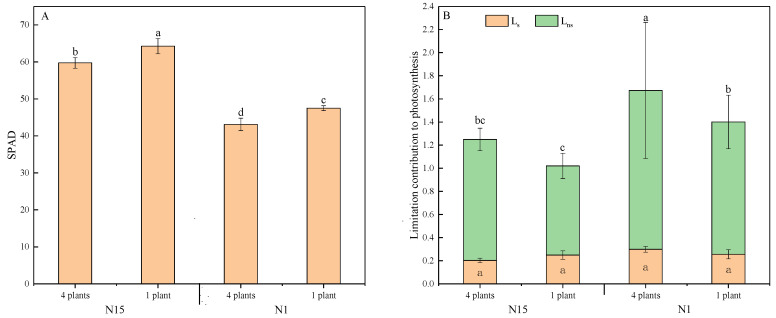
Chlorophyll index (SPAD) (**A**) and limitation contribution to photosynthesis (**B**) in response to neighbors in two nitrogen applications 14 days after transplanting. Mean values and SE (*n* = 6) are presented. Different letters indicate a significant difference at the *p* < 0.05 level by Duncan’s multiple-range test.

**Figure 6 plants-09-01674-f006:**
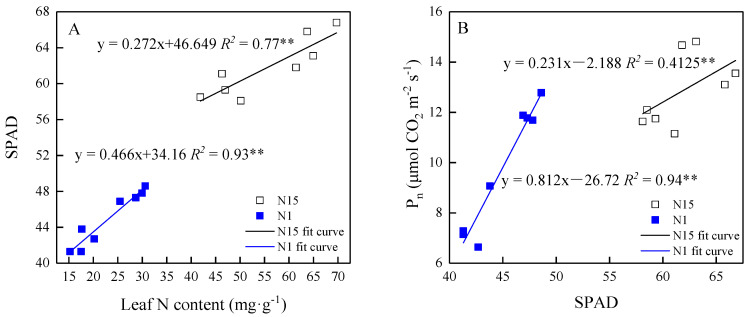
Relationships of chlorophyll index (SPAD) to leaf N concentration (**A**), P_n_ to SPAD (**B**), under N15 and N1 treatments. Open squares indicate plants under N15 treatment; closed squares indicate that plants under N1 treatment. Fit curve is presented with the fitted equation (*n* = 8), ** indicates the regression lines are statistically significant at the *p* < 0.01 level.

**Figure 7 plants-09-01674-f007:**
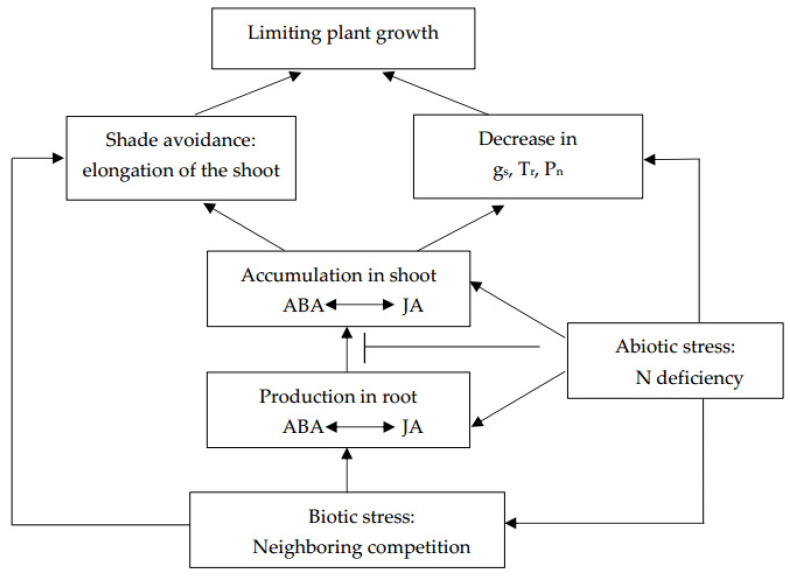
Proposed model of neighboring competition induced plant morphological and physiological response under N deficiency conditions. Lines ending in arrowheads indicate positive impacts, while lines ending in a bar indicate negative impacts.

**Table 1 plants-09-01674-t001:** Shoot length, stem diameter, leaf area and dry biomass of single and competing plants under different nitrogen conditions.

Treatment		Shoot Length /(cm)	Stem Diameter /(mm)	Leaf Area /(cm^2^·plant^−1^)	Dry Biomass /(g·plant^−1^)
N15P4		12.07 ± 0.35 ^a^	3.02 ± 0.09 ^b^	94.21 ± 7.58 ^b^	0.368 ± 0.016 ^b^
N15P1		12.24 ± 0.16 ^a^	3.21 ± 0.05 ^a^	138.23 ± 9.71 ^a^	0.598 ± 0.026 ^a^
N1P4		10.40 ± 0.45 ^c^	2.25 ± 0.03 ^d^	44.38 ± 3.70 ^d^	0.195 ± 0.010 ^d^
N1P1		11.21 ± 0.32 ^b^	2.61 ± 0.02 ^c^	69.69 ± 6.12 ^c^	0.261 ± 0.014 ^c^
Factors	*df*	*p*-value			
Competition (C)	1	0.002 ^**^	0.000 ^**^	0.000 ^**^	0.000 ^**^
Nitrate application (N)	1	0.000 ^**^	0.000 ^**^	0.000 ^**^	0.000 ^**^
C × N	1	0.029 ^*^	0.002 ^**^	0.001 ^**^	0.000 ^**^
Error	20				

Data represents mean ± SE. * and ** within a column indicate a significant difference between treatments at the *p* < 0.05 or *p* < 0.01 level by two-way ANOVA, Duncan’s multiple-range test, *n* = 6.

**Table 2 plants-09-01674-t002:** Effects of competition and nitrate application on concentrations of abscisic acid (ABA), jasmonic acid (JA) in shoots and roots by two-way variance analysis.

Factors	*df*	*p*-Value					
		Shoot ABA	Root ABA	ABA Ratio	Shoot JA	Root JA	JA Ratio
Competition (C)	1	0.000 **	0.000 **	0.010 *	0.000 **	0.018 *	0.008 **
Nitrate application (N)	1	0.959 ^ns^	0.000 **	0.000 **	0.000 **	0.002 **	0.000 **
C × N	1	0.010 **	0.000 **	0.001 **	0.000 **	0.000 **	0.000 **
Error	8						

* represents significant difference at the *p* < 0.05 level; ** represents significant difference at the *p* < 0.01 level; ns represents no significant difference between treatments by two-way ANOVA, Duncan’s multiple-range test, *n* = 3.

**Table 3 plants-09-01674-t003:** Relationships of g_s_ and P_n_ to ABA and JA concentrations in shoots, roots and shoot-to-root ratio by Pearson’s correlation coefficient analysis under different N applications.

Factors	g_s_ in N15	g_s_ in N1	P_n_ in N15	P_n_ in N1
shoot ABA	‒0.872 **	‒0.912 **	‒0.863 **	‒0.941 **
root ABA	‒0.936 **	‒0.966 **	‒0.821 **	‒0.980 **
ABA ratio	‒0.715 *	‒0.895 **	‒0.834 *	‒0.866 **
shoot JA	‒0.715 *	‒0.992 **	‒0.791 *	‒0.977 **
root JA	0.476	‒0.950 **	‒0.744 *	‒0.967 **
JA ratio	‒0.556	‒0.984 **	‒0.757 *	‒0.937 **

* represents significant difference at the *p* < 0.05 level; ** represents significant difference at the *p* < 0.01 level.

**Table 4 plants-09-01674-t004:** Effects of competition and nitrogen application on chlorophyll index (SPAD), the contribution of stomatal limitation (L_s_) and non-stomatal limitation (L_ns_) by two-way variance analysis.

Factors	*df*	*p*-Value		
		SPAD	L_s_	L_ns_
Competition (C)	1	0.000 ^**^	0.959 ^ns^	0.010 ^**^
Nitrate application (N)	1	0.000 ^**^	0.001 ^**^	0.000 ^**^
C × N	1	0.907 ^ns^	0.002 ^**^	0.258 ^*^
Error	20			

* represents significant difference at the *p* < 0.05 level; ** represents significant difference at the *p* < 0.01 level; ns represents no significant difference between treatments by wo-way ANOVA, Duncan’s multiple-range test, *n* = 6.

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
