# Peer review of "Leaf Gas Exchange of Tomato Depends on Abscisic Acid and Jasmonic Acid in Response to Neighboring Plants under Different Soil Nitrogen Regimes"

_plants, 2020, doi:10.3390/plants9121674_

Round 1

Reviewer 1 Report

The study is well designed and executed, provides novel data which have the potential to contribute to the field. I have only some small suggestions for improvement.

  • Could you please provide the GPS coordinates for the experimental settings?
  • Please, add P values in the description of significant results.
  • Because the Material and Methods section is placed following the Results and Discussion, please provide a description of the abbreviations used in the Results and Discussion.
  • A small language revision would be necessary, as some of the sentences are written in informal English. Please, do not start the sentences with "And".
  • Please, add limitations of the study as well as future prospects into the Discussion.

Author Response

Comment 1: Could you please provide the GPS coordinates for the experimental settings?

Response: The GPS coordinates were added to the section of Materials and Methods (Lines 316-317).

Comment 2: Please, add P values in the description of significant results.

Response: Thanks. P values were added in the Results section when descripted significant results (such as in lines 110, 121-122, 125-126, 138-150).

Comment 3: Because the Material and Methods section is placed following the Results and Discussion, please provide a description of the abbreviations used in the Results and Discussion.

Response: Thanks for your suggestions. A description of the abbreviations used in the Results and Discussion was added (such as in lines 119, 123-124, 190, 221).

Comment 4: A small language revision would be necessary, as some of the sentences are written in informal English. Please, do not start the sentences with "And".

Response: English of the manuscript was polished, and mistakes were corrected.

Comment 5: Please, add limitations of the study as well as future prospects into the Discussion.

Response: The limitations of the study was added (lines 251-253 and 285-287) and future prospects were also added (lines 266-268 and 393-394).

Reviewer 2 Report

The authors have studied the role of ABA, and JA and their orchestrated signaling in response to neighbor competition under different nitrogen availability conditions. 

The experiments are conducted well and the data is presented appropriately. However, the writing can be improved in many parts, in particular the Results and Discussion sections. Although the English usage is good, one could see many grammatical and typographical errors throughout the manuscript. Hence a thorough proofreading is highly recommended prior to the acceptance for publication.

Additionally, though the results have been discussed adequately, it would be better if the authors can provide a hypothetical scheme/illustration based on their findings, and discuss it in Discussion or Conclusion section. 

Similarly, the authors need to provide morphological differences (potted plants) in the plants under different N regime.

Some minor points:

L14: "....shortage are two important limiting......."

L27: What is R^2? Explain briefly. Note that not all of the readers are expert in statistical analyses and might not understand the meaning of these terms.

L28: compared to in? Sentence not clear.

L44: "....canopies usually receive low quality...."

L46: "....past studies have shown..."

L54: "....inoculated with producing cytokinin..."?? Did you mean cytokinin-producing bacteria?

L62: "...lower in unable accumulation..."? sentence not clear

L63: "....high production of JA..."

L66: "In addition, few studies have demonstrated......"

L67: "....need work synergistically...."?? Sentence not clear.

L70: "....as one of the most...."

L79: Solanum lycopersicum should be italicized

L79: "....one of the most important vegetable crops......"

L90: "....N1- treated plants. Two-way ANOVA results....". Do not start a sentence with "And".

Figure 1: Explain in the figure caption, what C:ns/N:ns/CxN:ns mean? 

L100" ".....could significantly impact the growth indexes (Table 1)".

Table 1: The letters showing significant differences should be in superscript. The P value should be mentioned in the caption.

L114: What is Ci? It needs to be described before.

L114: Ci increased in comparison to what?

Figure 2 and other figures: Letters indicating significant differences should be above the error bars.

L134: "....2.3 times higher shoot JA as compared to tat of single plant (Figure 3B)."

Figure 3: P value should be mentioned in the figure caption.

L149: Again, R^2 and its relevance needs to be clearly explained.

L153: "The absolute value....". Do not start a sentence with "And".

Figure 5: Letters showing significant difference should be above the error bars.

L211: ".....because more ABA accumulated in shoot than in root...."

L219: "deficient plants"

L221: ".....than that of N15-treated plants"

L228: The heading should be "Effect of neighbor competing and nitrogen availability on leaf gas exchange and JA"

L230: ".....roots were also investigated".

L230: "We found that accumulation of JA increased significantly in the shoots.........."

L232: "....The data presented here were in agreement with previous studies that showed increased JA accumulation under water stress".

L234: The sentence is not clear. Should be rephrased clearly.

L237-240: The sentence is not clear. Should be rephrased clearly.

L241: "....increasing shoot JA, indicating that....".

L246-249: Rephrase to make more clear.

L257: which plants?

L262: Heading should be "Effect of neighbor competing and nitrogen availability on Pn"

L267: "The N1-treated plants....". Delete "Moreover"

L273: ".....surprising that plant growth parameters such as.....".

L275: "....low nitrogen availability (Table 1)."

Author Response

Comment 1: The “Introduction” starts explaining shade avoidance and the consequent R/FR ratio alterations derived, goes on to refer to ABA/JA relations on the regulation of stomatal aperture regulation (stomatal conductance)and gas exchanges and then introduces the matter of soil nitrogen supply. It is understanding they make this reference since the authors object study is overcrowding growth conditions and indeed shade avoidance is a possible concern but the connection between this problematic and ABA/JA and then the added aspect of nitrogen supply does not make for a properly through flowing text. It gives the impression that they are somewhat separate blocks in the way they are described. Possibly authors should start by referring to overcrowding and nitrogen supply in relation to ABA/JA then refer to the shade avoidance phenomenon which could be relevant to the overcrowding situation under study and the effects this adaptation can have on ABA and JA levels or interactions and consequences for stomatal aperture (as measured by stomatal conductance) and then the interaction of overcrowding with nitrogen supply.

Response: Many Thanks for your suggestions. The first paragraph of Introduction was revised according to this suggestion. The Introduction starts by referring overcrowding and nitrogen firstly (Lines 42-46), and describes the effects of overcrowding and nitrogen supply on ABA (Lines 46-51), then refers to the shade avoidance phenomenon and the roles of plant hormones in response to this adaptation (Lines 51-75), and refers to the interaction of ABA and JA in response to overcrowding and nitrogen supply (Lines 75-78). 

Comment 2: The data presented is according to this reviewer interesting as it describes the combination between crowding and soil nitrogen supply as it regards stomatal behavior indicating regulation of H2O losses, gas exchanges, growth and the actions of two hormonal factors involved in this phenomenon, ABA and JA. This reviewer considers that the data is weaker in this respect because it lacks information on signaling processes in these conditions. The authors present data suggesting that there is accumulation of ABA in the shoots of high nitrogen crowded conditions and in the discussion they claim this is due to export from root to shoot. This evidence is indirect as there is no demonstration that this is the case basing their assumption on published data for other plant systems. Moreover on low nitrogen conditions the relation between levels of ABA in root to shoot were not changed. How much is the nitrogen levels in the soil affecting export and why?

Response: Though the origin of ABA is controversial, it is widely accepted that both root and leaf can produce ABA. Since Davies and Zhang (1991) have stressed the role of ABA as a long-distance hormonal signal in plants in drying soil. Vysotskaya et al (2017) used the spraying fluridone, an inhibitor of ABA transport, to competing plants growing by three per pot and showed there was no accumulation of ABA in shoot and inhibition of growth in competing plants (Lines 223-225). In addition, Li et al (2018) used the methods of grafting ABA-deficient mutant roots into wild-type tomato shoots and confirmed the opening of stomata was due to the attenuated root-to-shoot ABA transport (Lines 227-230). These studies clearly confirmed the importance of ABA as a long-distance hormonal signal in plants in stresses.

Reference: (1) Davies, W.J.a.Z.J. Root signals and the regulation of growth and development of plants in drying soil. Annu Rev Plant Biol 1991, 42, 55-76..

(2) Li, W.; de Ollas, C.; Dodd, I.C. Long-distance ABA transport can mediate distal tissue responses by affecting local ABA concentrations. J Integr Plant Biol 2018, 60, 16-33, doi:10.1111/jipb.12605.

(3)Vysotskaya, L.B.; Arkhipova, T.N.; Kudoyarova, G.R.; Veselov, S.Y. Dependence of growth inhibiting action of increased planting density on capacity of lettuce plants to synthesize ABA. J Plant Physiol 2018, 220, 69-73, doi:10.1016/j.jplph.2017.09.011.

Shoot to root ABA/JA ratio in N1 was lower than in N15, indicating N deficiency might decrease ABA transport from root to shoot (Figure 3A). This conclusion maybe partly due to the decreased Tr in N1. There are many studies reported that N availability play a major effect on leaf gas exchange, N deficiency commonly decrease Pn, gs and Tr (Lines 234-242). This was described in the reviewed manuscript as “One of the reasons might be due to the lower Tr in N1-treated plants (Figure 2C). Transpirational pull was considered to be the main power for plants to transport water, hormones and other nutrient from root [32]. N supply was expected to have a major effect on leaf gas exchange, particularly on gs and Tr [33, 34]. Another reason may be due to more ABA produced in root under N deficiency condition. The results of more ABA accumulated in N-deprived plants were in agreement with the data of Oka et al. [35], who showed that low nitrogen availability promoted ABA accumulation. N deficiency has been shown to upregulate abundance of zeaxanthin epoxidase (TaZEP), a key ABA synthesis-related enzyme and increase the endogenous ABA content [36].”.

Comment 3: The interaction between ABA and JA is discussed but it is not demonstrated clearly by the data presented in this ms.

Response: Thanks. This part was revised in the manuscript. ABA and JA in shoot of competing plants increased both in N15 and N1 condition, ABA and JA in root increased except in N1 condition. It was suggested that ABA and JA work synergistically instead of oppositely in response to the neighbors. So we deduced there might be a coordinated interaction between JA and ABA in response to the neighbors. However, we cannot fully explain the proposed conclusion in this experiment, we will use some specific plant materials (such as ABA/JA deficient mutant) and new methods (such as grafting different mutants) to study this interaction or signaling pathways.

Comment 4: This reviewer believes the text in the discussion needs to be made clearer and the English must be improved as this section is somewhat confusing and recursive.

Response: Thanks very much for your comments. The discussion section was reviewed again and English was also polished.

Reviewer 3 Report

The Ms from Li S et al. titled “Leaf gas exchange of tomato depends on Abscisic acid and Jasmonic acid in response to neighboring plants under different soil nitrogen regimes” presents physiological data on the interaction between nitrogen supply and crowding as well as the distribution of ABA and JA in these conditions.

The “Introduction” starts explaining shade avoidance and the consequent R/FR ratio alterations derived, goes on to refer to ABA/JA relations on the regulation of stomatal aperture regulation (stomatal conductance)and gas exchanges and then introduces the matter of soil nitrogen supply. It is understanding they make this reference since the authors object study is overcrowding growth conditions and indeed shade avoidance is a possible concern but the connection between this problematic and ABA/JA and then the added aspect of nitrogen supply does not make for a properly through flowing text. It gives the impression that they are somewhat separate blocks in the way they are described. Possibly authors should start by referring to overcrowding and nitrogen supply in relation to ABA/JA then refer to the shade avoidance phenomenon which could be relevant to the overcrowding situation under study and the effects this adaptation can have on ABA and JA levels or interactions and consequences for stomatal aperture (as measured by stomatal conductance) and then the interaction of overcrowding with nitrogen supply.

The data presented is according to this reviewer interesting as it describes the combination between crowding and soil nitrogen supply as it regards stomatal behaviour indicating regulation of H2O2 losses, gas exchanges, growth and the actions of two hormonal factors involved in this phenomenon, ABA and JA. This reviewer considers that the data is weaker in this respect because it lacks information on signalling processes in these conditions. The authors present data suggesting that there is accumulation of ABA in the shoots of low nitrogen crowded conditions and in the discussion they claim this is due to export from root to shoot. This evidence is indirect as there is no demonstration that this is the case basing their assumption on published data for other plant systems. Moreover on low nitrogen conditions the relation between levels of ABA in root to shoot were not changed. How much is the nitrogen levels in the soil affecting export and why?

The interaction between ABA and JA is discussed but it is not demonstrated clearly by the data presented in this ms.

This reviewer believes the text in the discussion needs to be made clearer and the English must be improved as this section  is somewhat confusing and recursive.

Author Response

Comment 1: The experiments are conducted well and the data is presented appropriately. However, the writing can be improved in many parts, in particular the Results and Discussion sections. Although the English usage is good, one could see many grammatical and typographical errors throughout the manuscript. Hence a thorough proofreading is highly recommended prior to the acceptance for publication.

Response: Thanks for your suggestions. English of the manuscript was improved and the mistakes were corrected.

Comment 2: Additionally, though the results have been discussed adequately, it would be better if the authors can provide a hypothetical scheme/illustration based on their findings, and discuss it in Discussion or Conclusion section.

Response: Thanks. We have added the hypothetical model of neighboring competition induced plant morphological and physiological response under N deficiency conditions to illustrate our findings (Figure 7, Lines 255-257).

Comment 3: Similarly, the authors need to provide morphological differences (potted plants) in the plants under different N regime.

Response: Thanks. We have provided the picture in a supplementary file, i.e. the supplementary material. The morphological differences between different treatments in the end of our experiment were shown in this figure.

All minor points have been revised following your suggestions.

  • L14: "....shortage are two important limiting......."

Response: “of” was deleted (Line 14).

  • L27: What is R2? Explain briefly. Note that not all of the readers are expert in statistical analyses and might not understand the meaning of these terms.

Response: Thanks. R2 is the Pearson's correlation coefficient. The explanation was added (Line 27).

  • L28: Compared to in? Sentence not clear.

Response: This sentence was revised as "As compared to the absolute value of slope of gs to shoot ABA in N15, it increased in N1 treatment." (Line 28-29).

  • L44: "....canopies usually receive low quality...."

 Response: "receive" replaced "received" (Line 52).

  • L46: "....past studies have shown..."

 Response: "shown" replaced "showed" (Line 53).

  • L54:"....inoculated with producing cytokinin..."?? Did you mean cytokinin-producing bacteria?

Response: Yes, it is a bacteria with capable of producing cytokinin. This sentence was corrected as "Moreover, Arkhipova et al. [14] found that lettuce plants inoculated with bacteria, Bacillus subtilis, strain IB-22, capable of producing cytokinin, did not reduce Tr in response to neighbors." (Lines 61-63).

  • L62: "...lower in unable accumulation..."?

Response: This sentence was corrected as "More recently, de Ollas et al. [22] also reported that stomatal sensitivity to drying was lower in def-1 self-grafted mutant (which fails to accumulate JA in shoot) plants as compared to the def-1 scions grafting with WT rootstocks." (Lines 69-71).

  • L63: "....high production of JA..."

 Response: " of " was added (Line 71).

  • L66:"In addition, few studies have demonstrated......"

Response: This sentence was rephrased as "In addition, few studies have demonstrated that the role of JA inhibiting stomatal opening need interplay with ABA, as JA was suggested to have a role in ABA biosynthesis [24,25]." (Lines 76-77).

  • L67: "....need work synergistically...."?? Sentence not clear.

Response: This sentence was revised as "In addition, few studies have demonstrated that the role of JA inhibiting stomatal opening need interplay with ABA, as JA was suggested to have a role in ABA biosynthesis [24,25]." (Lines 76-77).

  • L70: "....as one of the most...."

Response: "of" was added (Line 44).

  • L79: Solanum lycopersicum should be italicized.

Response: " Solanum lycopersicum" was italicized (Line 86).

  • L79: "....one of the most important vegetable crops......"

Response: "crops" was added (Line 86).

  • L90: "....N1- treated plants. Two-way ANOVA results....". Do not start a sentence with "And".

Response: The sentence was revised according to the suggestion (Line 97).

  • Figure 1: Explain in the figure caption, what C:ns/N:ns/CxN:ns mean?

Response: Thanks, the meanings has been explained in the Figure 1 (Lines 104-105).

  • L100" ".....could significantly impact the growth indexes (Table 1)".

Response: “the” was added (Line 108).

  • Table 1: The letters showing significant differences should be in superscript. The P value should be mentioned in the caption.

Response: Thanks, letters were shown in superscript in Table 1 and the P value was been added under the table.

  • L114: What is Ci? It needs to be described before.

Response: Ci is the intercellular CO2 concentration, which was added in this sentence (Lines 123-124).

  • L114: Ci increased in comparison to what?

Response: This sentence was rephrased as “Intercellular CO2 concentration (Ci) in competing plants significantly increased in comparison to single plant under N15 condition (P=0.022)” (Line 124).

  • Figure 2 and other figures: Letters indicating significant differences should be above the error bars.

Response: Thanks, the figures were revised according to the suggestion.

  • L134: "....2.3 times higher shoot JA as compared to that of single plant (Figure 3B)."

Response: Thanks, "that of" was added (Line 147).

  • Figure 3: P value should be mentioned in the figure caption.

Response: P value was under the Figure 3. The specific P value was in table 2.

  • L149: Again, R^2 and its relevance needs to be clearly explained.

Response: R2 is Pearson's correlation coefficient, the explanation was added (Line 163).

  • L153: "The absolute value....". Do not start a sentence with "And".

Response: This sentence was corrected according to suggestions and "And" was deleted (Line 168).

  • Figure 5: Letters showing significant difference should be above the error bars.

Response: Letters were put above the error bars in Figure 5.

  • L211: ".....because more ABA accumulated in shoot than in root...."

Response: Thanks for your kind suggestion, this sentence was rephrased as Previous studies have reported that competition-induced stomatal closure was associated with the accumulation of ABA in shoots due to the increased transport of ABA from root to shoot under the increased planting density [6,29,30](Lines 223-225).

  • L219: "deficient plants"

Response: "Deficiency" was replaced by "deficient" (Line 243).

  • L221: ".....than that of N15-treated plants"

Response: "in" was deleted (Line 245).

  • L228: The heading should be "Effect of neighbor competing and nitrogen availability on leaf gas exchange and JA"

Response: Heading was "3.2. Effects of neighbor competing and nitrogen availability on leaf gas exchange and JA" (Line 260).

  • L230: ".....roots were also investigated".

Response: "root" was changed into "roots" and this sentence was describe as "……, changes of JA in shoots and roots were also investigated in our experiment." (Lines 261-262).

  • L230: "We found that accumulation of JA increased significantly in the shoots.........."

Response: This sentence was rephrased as "We found that the concentrations of JA in shoots increased in competing plants (Figure 3B)." (Lines 262-263).

  • L232: "....The data presented here were in agreement with previous studies that showed increased JA accumulation under water stress".

Response: Thanks. This sentence was revised according to your suggestion and described as "The data presented here was in agreement with previous studies showing that the concentration of JA in shoots was able to increase in response to water stress [17, 39]." (Lines 263-265).

  • L234: The sentence is not clear. Should be rephrased clearly.

Response: This sentence was rephrased as "On the contrary, some studies showed that low R/FR ratio light, an early warning signal for crowding, would inhibit the expression of JA-induced gene and promote attack of disease symptoms in Arabidopsis [40] and tomato [41]." (Lines 271-273).

  • L237-240: The sentence is not clear. Should be rephrased clearly.

Response: This sentence was rephrased as "Though we did not measure the effect of JA on plant disease symptoms, different results might be responsible for different plant materials, cultivars and growth conditions." (Lines 273-275).

  • L241: "....increasing shoot JA, indicating that....".

Response: "indicating" was replaced "replacing" (Line 276).

  • L246-249: Rephrase to make more clear.

Response: This sentence was revised as "In addition, the sensitivity of gs to drying decreased in def-1 self-grafts (which fail to accumulate JA in shoot during drying) as compared to def-1 scions grafted onto WT rootstocks [19]." in lines 283-285.

  • L257: which plants?

Response: It is "rice cultivars " (Line 296).

  • L262: Heading should be "Effect of neighbor competing and nitrogen availability on Pn"

Response: Heading was described as "Effects of neighbor competing and nitrogen availability on Pn" (Line 302).

  • L267: "The N1-treated plants....". Delete "Moreover"

Response: "The" was added and "Moreover" was deleted (Line 307).

  • L273: ".....surprising that plant growth parameters such as.....".

Response: "parameters" was added (Line 314).

  • "....low nitrogen availability (Table 1)."

Response: "availability" was added (Line 315).
